# Male reproductive traits are differentially affected by dietary macronutrient balance but unrelated to adiposity

A. J. Crean [1], S. Afrin[1], H. Niranjan[1], T. J. Pulpitel[1], G. Ahmad[1,2], A. M. Senior [1], T. Freire[1], F. Mackay[1], M. A. Nobrega[3], R. Barrès [4,5], S. J. Simpson [1] & T. Pini [1,6] ✉

Dietary factors influence male reproductive function in both experimental and epidemiological studies. However, there are currently no specific dietary guidelines for male preconception health. Here, we use the Nutritional Geometry framework to examine the effects of dietary macronutrient balance on reproductive traits in C57BL/6 J male mice. Dietary effects are observed in a range of morphological, testicular and spermatozoa traits, although the relative influence of protein, fat, carbohydrate, and their interactions differ depending on the trait being examined. Interestingly, dietary fat has a positive influence on sperm motility and antioxidant capacity, differing to typical high fat diet studies where calorie content is not controlled for. Moreover, body adiposity is not significantly correlated with any of the reproductive traits measured in this study. These results demonstrate the importance of macronutrient balance and calorie intake on reproductive function and support the need to develop specific, targeted, preconception dietary guidelines for males.

Nutrition is one of the most important environmental factors influencing health and fertility[1]. Most studies investigating dietary impacts on male reproduction have focussed on impacts of excess body fat, likely due to the increasing prevalence of obesity in males of reproductive age[2–4]. Multiple mechanisms link obesity with male reproductive function, including hormonal imbalance, inflammation, oxidative stress, and increased scrotal temperature (reviewed in refs. 5,6). However, it is not clear whether impaired reproductive function in males with obesity is the result of increased adiposity, intake of specific nutrients, or the indirect effect of health complications associated with obesity. Interestingly, a recent meta-analysis concluded that bariatric surgery does not improve semen parameters[7]. This suggests that body weight loss is not sufficient to ameliorate male infertility, and thus preconception dietary advice in males needs to move beyond the sole directive of maintaining a healthy weight. To develop specific dietary guidelines, there is a need to better understand the relative influence of different dietary components on reproductive function, independent of effects of energy intake.

Animal models testing for effects of diet on male fertility typically use a high-fat or Western versus control diet. These models consistently show that rodents fed a high-fat diet have reduced sperm quality compared to rodents fed a control chow diet[8]. However, there are typically several differences between the experimental diets, including energy density and macronutrient ratios. Therefore, it is not clear which dietary factor (or combination of factors) causes the observed decline in sperm quality (reviewed in[9]). Nutritional Ecology and Nutritional Geometry provide an experimental framework for disentangling the effects of energy content and the relative proportions of dietary components on health traits of interest[10]. This approach has demonstrated that dietary macronutrient balance, not

[1]Charles Perkins Centre and School of Life and Environmental Sciences, The University of Sydney, Camperdown, NSW 2006, Australia. [2]Department of Andrology, Royal Women's and Children's Pathology, Royal Women's Hospital, Parkville, VIC 3053, Australia. [3]Department of Human Genetics, University of Chicago, Chicago, IL 60637, USA. [4]Novo Nordisk Foundation Center for Basic Metabolic Research, University of Copenhagen, Copenhagen DK-2200, Denmark. [5]Institut de Pharmacologie Moléculaire et Cellulaire, Université Côte d'Azur & Centre National pour la Recherche Scientifique (CNRS), Valbonne 06560, France. [6]School of Veterinary Science, The University of Queensland, Gatton, QLD 4343, Australia. ✉e-mail: t.pini@uq.edu.au

calories consumed alone, affects health and rate of aging in ad libitum-fed mice[11]. Nutritional Geometry has also shown that optimal dietary macronutrient ratios change across the life-course, in response to environmental circumstances, and between different life-history traits[12,13]. For example, lifespan is maximised on a low-protein/high-carbohydrate diet, whereas reproduction is generally optimised by a diet with higher protein content[14–16].

In addition to macronutrient balance, macronutrient quality can significantly alter health outcomes and is an important factor to consider when designing dietary studies. The balance of amino acids affects growth, reproduction, cardiometabolic health and ageing[17]. High-fat/Western diets usually contain excessive omega-6 fatty acids yet are deficient in omega-3 fatty acids[18]. Elevated saturated and trans-fatty acids (as found in Western diets) are negatively correlated with sperm quality[19,20] and positively correlated with testicular histological abnormalities, oxidative stress and apoptosis[21]. Western diets also often contain excess sugars, with controversial effects on health which may be resolved by considering the effects of carbohydrate quality within the context of overall dietary macronutrient balance[22,23].

Here, we use the Nutritional Geometry framework to examine the effects of macronutrient balance on a range of outcomes related to reproduction and antioxidant function in C57BL/6J male mice. Importantly, the same ingredients were used to prepare each of the 10, isocaloric treatment diets; the only difference among diets was the proportion of each ingredient used, which was varied systematically across a physiologically relevant nutritional range. In summary, we explored the effect of macronutrient balance while controlling dietary energy density and macronutrient quality. This experimental design allowed us to visualise the impact of varying the proportion of dietary macronutrients on male reproductive function, providing a basis for the development of preconception dietary guidelines for males.

## Results

### Body composition and food intake

Diet had significant impacts on body composition (Fig. 1). Body weights were comparable at baseline, however diet significantly impacted body weight at sacrifice, weight change from baseline to sacrifice and average weekly weight gain. For all weight measures, low-protein diets resulted in the lowest values (diets 1, 2 mean weight at sacrifice $25.1 \pm 0.6$ g) and moderate protein, moderate carbohydrate, moderate fat diets resulted in the highest values (diets 3, 7 mean weight at sacrifice $33.1 \pm 0.5$ g). Although body weight had complex interactions with all macronutrients, it was significantly positively correlated with dietary protein ($r_s = 0.47$, $p = 8.3^{-8}$).

Body composition measured by quantitative magnetic resonance was significantly altered by diet (Fig. 1B, C). Lean mass was significantly driven by interactions between all macronutrients, and specifically had a significant moderate and positive correlation with dietary protein ($r_s = 0.53$, $p = 3.8^{-10}$). Fat mass was driven by interactions between fat and non-fat components, with the lowest fat mass produced by low protein, high fat diets and the highest fat mass resulting from a moderate fat, low protein, high carbohydrate diet. Analysing body fat as a percentage of total weight showed significant effects of protein and carbohydrate, with a low protein, high carbohydrate diet predicted to result in the highest body fat percentage. Correlation between body composition and reproductive parameters is presented in Fig. 2.

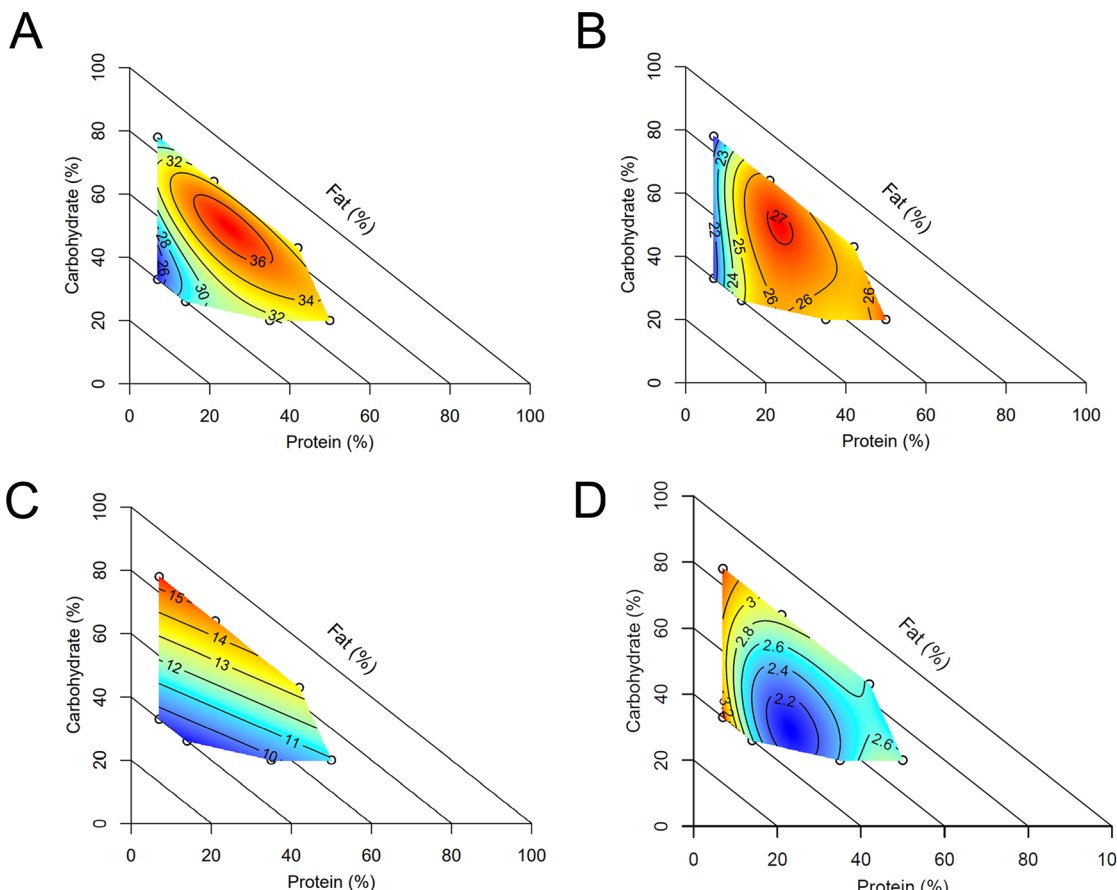

**Fig. 1 | Impact of diet on body composition and food intake.** Diet impacted body composition, measured by echoMRI including **A** body weight at sacrifice (g), **B** lean mass (g) and **C** percentage body fat (%), n = 12 males per diet. **D** Food intake (grams per day) differed significantly by diet, n = 6 males per diet. To interpret graphs; colour scale indicates the level of response (blue = minimum, red = maximum) with isolines showing the model predicted response. The diagonal axis is included to aid interpretation, depicting % dietary fat increasing from 0% at the outer perimeter to 100% at the origin. Source data are provided as a Source Data file.

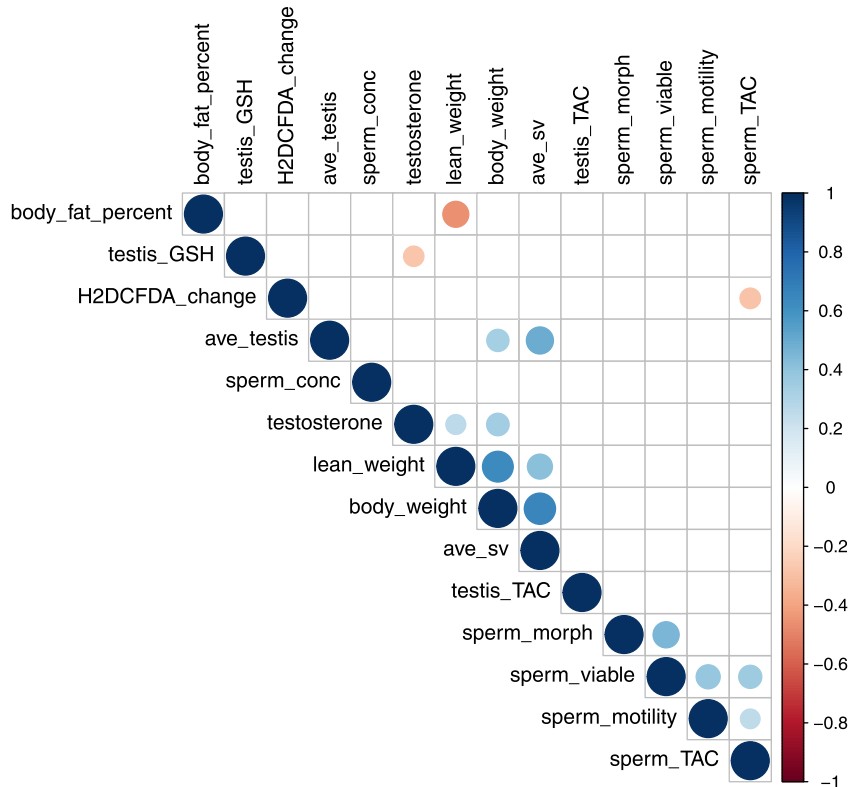

**Fig. 2 | Correlation of outcome variables.** Correlogram of body composition and reproductive parameters, with circle size and colour indicating correlation strength (Spearman's rho). Correlation was assessed by Spearman's rank correlation with two-sided hypothesis testing, with no adjustment for multiple comparisons. Presence of a circle indicates a significant ($p < 0.05$) Spearman correlation. $n = 6$ males per diet. Source data are provided as a Source Data file.

Food intake was significantly impacted by diet, with low protein diets resulting in higher intake (Fig. 1D). Overall, low protein diets with low carbohydrate and high fat tended to be associated with smaller males with a lower percentage body fat. In contrast, diets which were moderate in protein, fat and carbohydrate led to heavier males with a higher percentage body fat.

## Reproductive organ weights

Testis weight (average of left and right) was not significantly altered by diet (overall mean $97.6 \pm 1.0$ mg), however relative testis weight (average per gram of body weight) was significantly higher in males on a low protein diet, particularly at the extremes of high fat, low carbohydrate or low fat, high carbohydrate (Fig. 3A). Body weight was weakly correlated to testis weight (Fig. 3C, $r_s = 0.26$, $p = 0.0047$). In contrast, seminal vesicle weight (average of left and right) was significantly altered by diet, with a significant reduction in size in males on low P:C, high fat diets (diets 1 and 4; Fig. 3B). Seminal vesicle weight was moderately correlated with body weight (Fig. 3D, $r_s = 0.65$, $p = 8.8^{-16}$) and relative seminal vesicle weight (average per gram of body weight) showed a significant positive association with dietary protein (Table 1).

## Basic semen parameters

Epididymal sperm concentration was not significantly influenced by any of the diets (overall mean $45.0 \pm 1.2 \times 10^6$ spermatozoa/mL). Total motility was significantly impacted by diet composition, with a modest positive influence of dietary fat ($62.8 \pm 1.5\%$ motility in 15% fat diets compared to $69.2 \pm 2.1\%$ motility in 60% fat diets; Fig. 4A). Sperm morphology was significantly impacted by diet, and diets with moderate protein, regardless of fat content, tended to reduce abnormal morphology (Fig. 4B).

## Sperm viability and ROS production

There were no significant differences in sperm viability measured by LIVE/DEAD viability staining (overall mean $42.5 \pm 1.2\%$), although viability had a significant but weak correlation with total motility ($r_s = 0.39$, $p = 0.002$). While baseline $H_2DCFDA$ median fluorescence had significant, weak to moderate correlations with motility ($r_s = -0.27$, $p = 0.04$), viability ($r_s = -0.29$, $p = 0.03$) and abnormal morphology ($r_s = -0.52$, $p = 4.4^{-5}$), diet did not significantly impact baseline, or change from baseline, $H_2DCFDA$ median fluorescence.

## Histological analysis of spermatogenesis

Johnsen score was not significantly altered by diet, with a high mean score ($8.9 \pm 0.05$) indicating largely normal, active spermatogenesis across all treatments. However, the number of seminiferous tubules per section, the lumen diameter (short axis) and the degree of vacuolation were significantly impacted by diet composition (Fig. 5). The number of seminiferous tubules was reduced by moderate to high protein content and moderate fat content (diets 7 and 10). The average cross-sectional area of seminiferous tubules showed the inverse pattern and was negatively correlated with tubule number ($r_s = -0.39$, $p = 0.002$). Vacuolation within the seminiferous epithelium was increased either by high fat or high protein, and minimised by a low protein, high carbohydrate diet (diet 2). While measures of seminiferous tubule diameter and epithelial height were not significant, tubule lumen diameter (short axis) was significantly impacted by dietary C:F ratio, with low C:F leading to a substantially increased luminal diameter.

## Endocrine analysis

E2 was below the limit of detection in 56/60 samples, and neither E1 nor DHEA were detected in any samples. Testosterone (overall mean

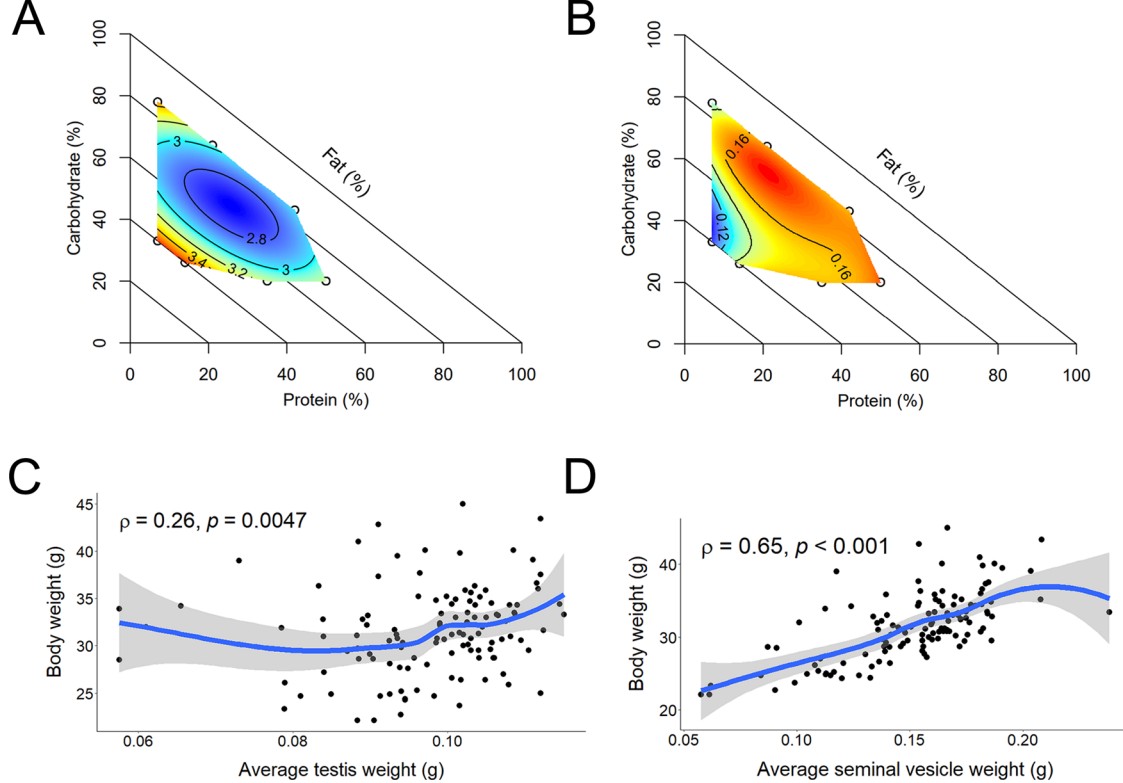

**Fig. 3 | Effects of diet on male reproductive organs.** Diet impacted reproductive organ weights, including **A** relative testis weight (mg/g body weight) and **B** average seminal vesicle weight. Correlation was assessed by Spearman's rank correlation with two-sided hypothesis testing, with no adjustment for multiple comparisons. Body weight was weakly related to **C** average testis weight ($r_s = 0.26$, $p = 0.0047$), but was more strongly correlated with **D** average seminal vesicle weight ($r_s = 0.65$, $p = 8.8^{-16}$). A trendline is shown in blue, with the shaded area indicating the 95% confidence interval. $n = 12$ males per diet. Source data are provided as a Source Data file.

4.03 ± 0.79 ng/mL, range 0.13 – 22.8 ng/mL) and DHT (overall mean 0.41 ± 0.07 ng/mL, range 0.05 – 3.28 ng/mL) concentrations varied widely between males but were not significantly different across diets (Supplementary data 2). There were no significant differences in 3α-diol or 3β-diol across diets, although 3β-diol was not detected in 14/60 samples (Supplementary data 2).

### Testicular gene expression
Testicular expression of steroidogenic acute regularly protein (*Star*) was not significantly affected by diet, however the expression of hydroxysteroid 17-beta dehydrogenase 3 (*Hsd17b3*) increased significantly with dietary protein (Fig. 6A).

Both glutathione peroxidases (*Gpx1* and *Gpx4*) were significantly impacted by diet, showing a positive relationship with dietary protein (Fig. 7A, B) and significant correlation with one another (Fig. 7D, $r_s = 0.66$, $p = 1.1^{-7}$). While glutathione reductase (*Gsr*) was not significantly impacted by diet, glutathione synthetase (*Gss*) showed increasing expression with a decreasing P:F ratio (Fig. 7C). While testicular expression levels of most antioxidant enzymes, including *Nfe2l2, Adh5, Sod1, Sod2, Sod3* and *Cat*, were not significantly altered by diet, expression of most antioxidant enzymes had significant, strong correlations (Fig. 7D).

### Testicular 8-OHdG DNA modification
There were no significant effects of diet detected on the abundance of 8-OHdG modifications in testicular DNA (overall mean 21.4 ± 1.1 ng/mL).

### Antioxidant activity of sperm and testes
Total antioxidant capacity of both testes and sperm was significantly impacted by diet. In the testes, total antioxidant capacity was primarily driven by a negative effect of protein, with the highest Trolox equivalent observed in males on low protein diets (Fig. 6B). In sperm, total antioxidant capacity showed a more complex pattern, with the lowest antioxidant concentration in low fat, moderate-high protein diets and the highest in high fat diets, at extremes of low and high dietary protein (Fig. 4C). Despite changes in overall antioxidant capacity, there was no significant difference in total glutathione concentration in the testes (overall mean 356.0 ± 14.4 μg/mL).

### Discussion
Here we show that male reproductive physiology is influenced by dietary macronutrient ratios (Table 1), and that the obesogenic nature of a diet does not appear to be a determining factor. Many studies investigating diet induced obesity have indicated that male reproductive parameters, including semen quality, testis architecture, gene expression and mating success are altered in individuals with obesity[9]. Interestingly, we found that percentage body fat was not highly correlated to any measure other than lean mass, demonstrating that aside from obesity, the composition of the diet can also precipitate significant changes in male reproductive function. This is generally supported by human observational studies linking dietary patterns with differences in semen quality in men with widely varying BMI[24, 25].

Using Nutritional Geometry, we were able to disentangle the effect of dietary fat separately from the effect of energy density. Traditional high fat, high calorie diets significantly reduce sperm motility[26–29]. In comparison, human studies categorising men by dietary fat intake found no significant correlation between fat intake and sperm motility[19,30–33]. We found a small but significant increase in sperm motility with increasing proportions of dietary fat. While not typically the primary energy substrate, β oxidation of fatty acids is an important metabolic pathway supporting sperm motility[34]. Further,

supplementation of particular fatty acids can increase sperm motility[35], even when paired with a classic high fat diet[36]. In our study, the ratio of omega-3 to omega-6 fatty acids was maintained at a ratio of 1: 3.7 and saturated fatty acids accounted for 23.2% of total fat content in all diets, whereas Western diets typically contain excessive amounts of saturated and trans-fatty acids and a deficiency of omega-3 fatty acids[18]. These differences in results demonstrate the importance of considering the source and balance of dietary fats in nutrition studies.

Alongside motility, sperm antioxidant capacity increased with the proportion of dietary fat. Low sperm antioxidant capacity is associated with infertility in men[37], suggesting that increased antioxidant activity may be beneficial for fertility. Mouse models have shown mixed results, including no difference[28], decreases[27,38], and increases[39] in sperm or testicular antioxidant activity under high fat diets. However, markers of oxidative stress are consistently elevated in these models[26,27,38–40]. In comparison, we found no evidence of oxidative stress by $H_2DCFDA$ staining and DNA 8-OHdG analyses in sperm and testes respectively, despite diets containing up to 60% of calories from fat. Elevated saturated and trans-fatty acids (as found in Western diets) are positively correlated with testicular histological abnormalities, oxidative stress, and apoptosis[21]. Hence, again, our contrasting results may reflect differences in dietary fatty acid balance compared to other studies.

In comparison to sperm, testicular total antioxidant capacity showed a negative relationship with dietary protein. However, despite changes in antioxidant activity, we did not observe depletion of the intracellular glutathione pool as others have[41]. It is unclear whether oxidative stress was induced and resolved by changes in antioxidant activity, or simply not produced by our dietary treatments. Similarly to high fat diets, low protein diets are associated with highly variable changes in antioxidant activity, but consistently elevated markers of oxidative stress in the testes[42] and other tissues[41,43]. Interestingly, a very high protein diet produced a similar effect[44]. However, given the considerable variability in antioxidant response to a low protein diet across tissues[41], the overall effect remains unclear. While previous data generally suggest a U-shaped effect curve between protein intake and antioxidant response, our results suggest that over a broad range of macronutrient ratios, there is a negative, linear relationship between dietary protein and antioxidant activity in the testis. These contradictory results may reflect differences in protein quality among studies. In our study, amino acid balance was optimised by matching proportions to the exome; i.e., calculating the relative proportion of each amino acid from a translation of genomic protein coding genes[45]. Flies fed an exome-matched low protein diet develop faster and lay more eggs than flies fed an equivalent yeast-based, exome mismatched diet[45,46]. Hence, we speculate that the high antioxidant capacity observed in low protein diets in our study is largely influenced by exome matching.

It is worth noting that while testicular antioxidant functional activity had a negative relationship with dietary protein, antioxidant gene expression did not. For example, glutathione peroxidases *GPx1* and *GPx4* showed similar increases in expression with increasing dietary protein, while glutathione synthetase (*Gss*) expression instead increased with dietary fat. While expression of most genes was not

## Table 1 | Summary of results

| Trait | Main dietary effect | Model favoured by AIC |
|---|---|---|
| **Body composition and reproductive organ weights** | | |
| Body weight at sacrifice | +ve Protein/–ve Fat | 4 |
| Lean mass | +ve Protein | 4 |
| Fat mass | +ve Carb/–ve Fat | 3 |
| % Body fat | +ve Carb/–ve Fat | 2 |
| Testis weight | – | 1 |
| Relative testis weight | +ve Fat/–ve Protein | 3 |
| Seminal vesicle weight | +ve Protein | 4 |
| Relative seminal vesicle weight | +ve Protein | 2 |
| **Semen parameters** | | |
| Sperm concentration | – | 1 |
| Sperm motility | +ve Fat | 2 |
| Sperm morphology | U Protein | 4 |
| Sperm viability | – | 1 |
| Sperm ROS production ($H_2DCFDA$) | – | 1 |
| Sperm total antioxidant capacity | +ve Fat | 3 |
| **Testis histology** | | |
| Johnsen score | – | 1 |
| # Seminiferous tubules (ST) | U Carb | 5 |
| ST area | – | 5 |
| ST lumen diameter | –ve Carb | 2 |
| ST epithelial height | – | 1 |
| ST epithelium vacuolation | –ve Carb | 3 |
| **Testicular RNA and DNA** | | |
| *Gpx1* & *Gpx4* | +ve Protein | 2 |
| *Gss* | +ve Fat | 2 |
| 8-OHdG | – | 1 |
| *Hsd17b3* | +ve Protein | 2 |
| Testis total antioxidant capacity | –ve Protein | 2 |

wct 3Mixture models fitted to data increasing in complexity from a null model (Model 1) through linear (Model 2), quadratic (Model 3) and cubic (Model 4) effects of protein, fat and carbohydrates on the trait of interest.

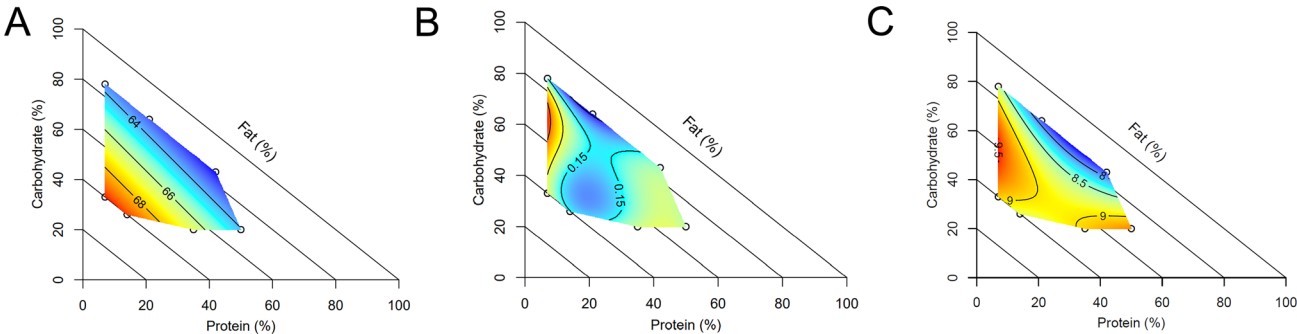

**Fig. 4 | Effect of diet on sperm quality.** Diet impacted sperm traits, including **A** total motility (%), **B** abnormal sperm morphology (proportion) and **C** sperm total antioxidant capacity (nmol Trolox equivalent/well). *n* = 6 males per diet. Source data are provided as a Source Data file.

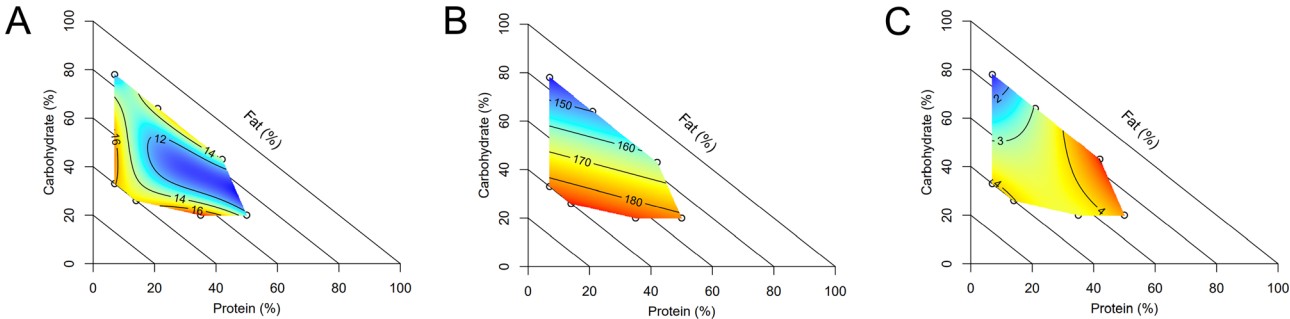

**Fig. 5 | Effect of diet on the microanatomy of the testis.** Diet impacted testicular architecture, including **A** the number of seminiferous tubules within a section, **B** average tubule lumen diameter (short axis, pixels) and **C** seminiferous epithelium vacuolation score. *n* = 6 males per diet. Source data are provided as a Source Data file.

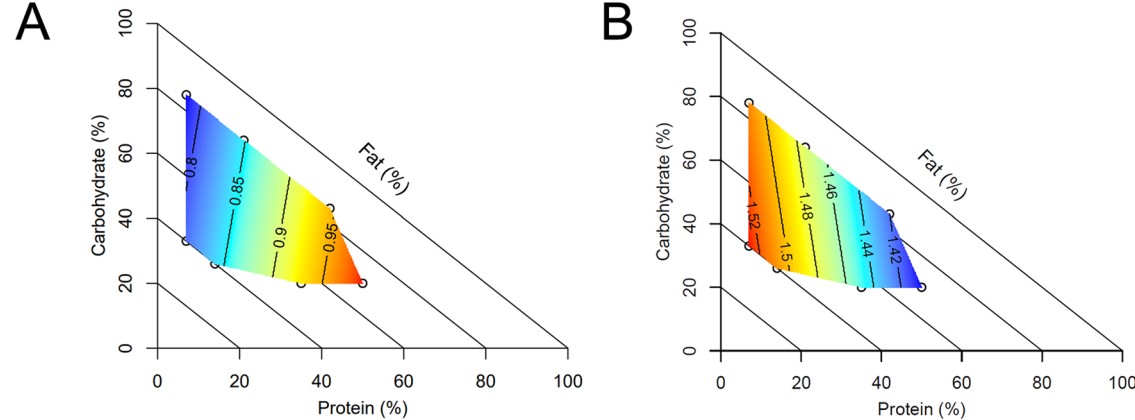

**Fig. 6 | Effect of diet on testosterone biosynthesis and antioxidant capacity.** Diet impacted testicular **A** expression of *Hsd17b3* and **B** total antioxidant capacity (nmol Trolox equivalent/μg protein). *n* = 6 males per diet. Source data are provided as a Source Data file.

significantly different across diets, expression levels were highly correlated. This is likely due to common upstream regulation by the *Nfe2l2* gene product Nrf2[47]. The disparity between enzymatic activity and expression is not necessarily surprising, as the two may not always be correlated[48,49]. Moreover, we measured total rather than enzyme specific antioxidant capacities. Overall, our results suggest that while diet can influence the expression of some antioxidants and overall antioxidant capacity, calorie intake and downstream effects of obesity are likely to have more profound effects on the generation of oxidative stress in the male reproductive tract.

Our investigation also highlighted additional important impacts of dietary protein on male reproductive function. Dietary protein had a positive effect on relative seminal vesicle mass, indicating effects of protein over and above the effects seen on body mass. Similar impacts of low protein diets on accessory sex gland size have been found in previous studies[15,50]. Western diets have been shown to alter seminal plasma composition[51,52], suggesting important effects of diet on glandular function. The reason for these observed differences is unclear but may be linked to changes in androgen concentration[53]. While an important testosterone biosynthetic enzyme (*Hsd17β3*) had lower testicular expression in low protein diets (as previously observed in bulls[54]), another enzyme in this pathway (*Star*) did not. Overall, we and others[26,28,33,40,55] found that serum testosterone was not significantly impacted by diet. This is potentially due to the pulsatile nature of testosterone secretion, leading to high variability in measured concentrations[56]. Interestingly, *Hsd17β3* is the rate limiting enzyme for maximal testosterone concentration[57], thus future studies should endeavour to determine the impact of diet on maximal androgen secretion.

Overall, mice across a broad range of diet compositions maintained normal spermatogenesis, producing similar numbers of viable mature sperm. As mice on low protein diets were smaller, the consistency in testis weight across diets means that mice on low protein diets invested relatively more in testis mass. Solon-Biet, et al.[15] found that testis mass was related to protein intake at 15 months of age in mice fed one of 25 experimental diets since weaning, suggesting that the overinvestment in testes relative to body mass of males fed low protein diets cannot be maintained into later age. Interestingly, despite the lack of effect on testis mass and epididymal sperm concentration, we did observe differences in testicular structure. Small numbers of vacuoles were found in the seminiferous epithelium of mice on diets with 42–50% of calories from protein. This has previously only been reported as an effect of high fat diets[21,58,59], in combination with severe spermatogenic disruption. Vacuolation of Sertoli cells is a common toxicant-induced injury[60], potentially suggesting a negative effect of excessive dietary protein. In contrast, more sperm with abnormal morphology were observed in mice on low protein diets (7% of calories) with high carbohydrate and low fat. Similar findings have been reported in other protein deficiency models[42]. Taken together, these results suggest that extremes of low and high protein consumption may have important impacts on seminiferous epithelium function.

It is legitimate to question the translation potential of our results to humans. Even compared to fertile humans, reproductive efficiency (successful pregnancy per cycle) is notably higher in mice (C57BL/6 84% vs human 42%)[61,62]. Given the overall poorer reproductive parameters in humans, as well as the multifaceted lifestyle and environmental factors negatively impacting male fertility[1], we postulate that even the modest effects which we observed may be more exacerbated in humans. Yet, it is worth reporting that none of the males showed complete infertility (i.e. inability to sire any offspring after multiple matings). However, the method of mating used was not an accurate,

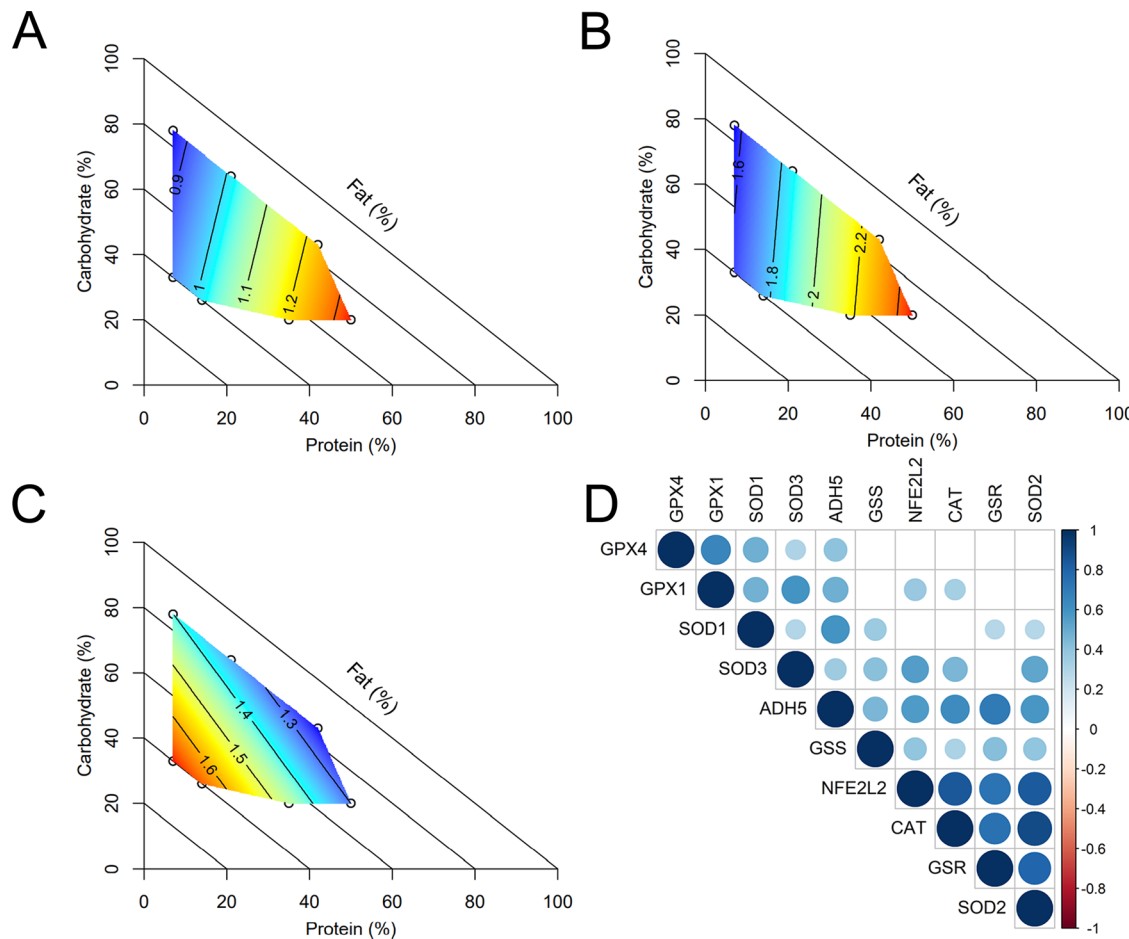

**Fig. 7 | Effect of diet on expression of antioxidant enzymes.** Diet impacted testicular antioxidant function, including relative expression of **A** *Gpx1*, **B** *Gpx4* and **C** *Gss*. **D** Correlogram of relative testicular expression of antioxidant genes, with circle size and colour indicating correlation strength (Spearman's rho). Correlation was assessed by Spearman's rank correlation with two-sided hypothesis testing, with no adjustment for multiple comparisons. Presence of a circle indicates a significant ($p < 0.05$) Spearman correlation. $n = 6$ males per diet. Source data are provided as a Source Data file.

controlled means of assessing reproductive efficiency. A more robust approach would be required to detect any subtle differences in fertility if they do exist. While in humans clearly defined cut offs correlated to fertility outcomes are used to predict male fertility[63], no such cut offs exist for mice. In using mice as a model, we have established that some of these parameters (e.g. sperm motility, morphology) can be impacted by diet. In men, these impacts could possibly bring individuals below the clinically accepted normal cut offs, however this remains to be investigated.

While this study has generated important insights into the impacts of diet on male reproductive physiology, it does have several limitations. As the diets were designed to be isocaloric, they consequently contained different amounts of indigestible cellulose. In particular, larger amounts of cellulose were required to dilute the energy content of high fat diets. While cellulose has negligible nutritional value, as a source of insoluble fibre it could potentially impact results. For instance, diets higher in insoluble fibre have been shown to increase sperm motility and alter gut microbiota in boars[64], although this result may have been influenced by concurrent changes in soluble dietary fibre. Overall, we felt that it was more important in this investigation to control for the effects of calories rather than the level of insoluble dietary fibre. An additional limitation of using the isocaloric diet approach is differences in food intake. Food intake did differ between diets as shown in the results and as previously reported[11]. Thus while data here is presented relative to the diet structure, these values do not necessarily match the actual amounts of each

macronutrient consumed by mice on different diets. Given the complexity of the Nutritional Geometry experimental design, it is most suited to comprehensive first pass studies, with simpler follow up experiments exploring selected diets of interest in greater detail.

The aim of this study was to examine the effects of diet on reproduction in a mouse model, with a view to eventual translation to human health. To do this, our experimental design covered wide macronutrient ranges (P 7–50%, C 20–78%, F 15–60%). While these fall outside the recommended Acceptable Macronutrient Distribution Range (AMDR; P 15–25%, C 45–65%, F 20–35%)[65] and observed human intake ranges (P 10–21%, C 35–62%, F 24–42%)[66], this design allowed us to clearly demonstrate both linear and more complex relationships that would not otherwise be apparent in a pairwise comparison (e.g. low fat vs high fat), or a design that was more limited in its coverage of nutrient space. It is worth noting however, that some outcomes were only significant in extreme ranges (e.g. seminiferous epithelium vacuolation on very high protein diets) and thus are less likely to be relevant in the human context.

An important aspect of this study was to further examine the links between diet, reproduction and obesity. Unfortunately, there is no widely accepted definition of obesity in mouse models, and the vast majority of studies consider a significant increase in body mass or weight change relative to the control as evidence of obesity[67]. While generating an obese mouse model was not the aim of this study, some of the diets clearly had obesogenic effects with maximal body fat of 33.9%, comparable to mice on a western diet[68]. Interestingly however,

body fat was not a good predictor for reproductive outcomes. This suggests that the links between diet, obesity and reproduction are more complex than originally thought, as effects originally categorised as resulting from obesity alone may instead be due to diet, or a combination of both factors. While diet and obesity can have a causal relationship, our results suggest that they should be considered independently as factors impacting reproduction.

While one macronutrient was the dominant effect in many cases, some parameters clearly had complex, non-linear relationships with multiple macronutrients. This reinforces the multifaceted physiological effects of different nutrients, and the importance of differentiating between the effects of diet, calories and body fat on reproductive function. Our findings clearly demonstrate that no single diet composition is beneficial for all sperm traits, testicular function or antioxidant capacity. Such uncoupling is important to consider in the development of guidelines aimed at improving male reproductive function prior to conception. It seems unlikely that a one size fits all approach will be beneficial for male fertility across the board. Instead, targeted research is required to better understand how diet could be used to improve outcomes in specific contexts such as asthenozoospermia, advanced paternal age and obesity, where improvement of one or several reproductive traits may be an effective treatment.

## Methods

### Chemicals
All chemicals were purchased from Millipore Sigma (Castle Hill, Australia) unless otherwise specified.

### Diets
Ten custom semi-pure experimental diets were produced by Specialty Feeds (Glen Forrest, Australia), based on AIN-93G. Diets were iso-caloric, with a net metabolisable energy of 14.7 MJ/kg (3.5 kcal/g) achieved using cellulose (indigestible fibre) to dilute energy as required. Protein content was exome-matched to the *Mus musculus* genome[45] by mixing casein and whey protein isolates supplemented with leucine, threonine, methionine, tyrosine, phenylalanine, tryptophan, alanine, aspartic acid, arginine, glycine, histidine and serine. Lipid sources included soybean oil, linseed oil and lard, with an omega 3 to omega 6 fatty acid ratio of 1:3.7 and saturated fats making up 23.2% of dietary fats. Carbohydrate sources included wheat starch, dextrinised starch and sucrose at a ratio of 4: 1.3: 1. Micronutrient content was kept consistent between diets. A breakdown of each diet by percentage of weight and percentage of net metabolisable energy contributed by protein, fat and carbohydrate are given in Table 2, and the full ingredient list and calculated nutritional parameters of each diet are provided in Supplementary file 1.

**Table 2 | Composition of experimental diets, with macronutrient content by weight and by metabolisable energy**

| Diet | % total weight[a] | | | % metabolisable energy | | |
|---|---|---|---|---|---|---|
| | Protein | Carbohydrate | Fat | Protein | Carbohydrate | Fat |
| **1** | 6.7 | 30.2 | 24.0 | 7.0 | 33.0 | 60.0 |
| **2** | 6.7 | 71.3 | 6.0 | 7.0 | 78.0 | 15.0 |
| **3** | 13.5 | 51.3 | 12.0 | 14.0 | 56.0 | 30.0 |
| **4** | 13.5 | 23.8 | 24.0 | 14.0 | 26.0 | 60.0 |
| **5** | 20.2 | 58.6 | 6.0 | 21.0 | 64.0 | 15.0 |
| **6** | 20.2 | 31.2 | 18.0 | 21.0 | 34.0 | 45.0 |
| **7** | 28.9 | 36.7 | 12.0 | 30.0 | 40.0 | 30.0 |
| **8** | 33.7 | 18.4 | 18.0 | 35.0 | 20.0 | 45.0 |
| **9** | 40.4 | 39.5 | 6.0 | 42.0 | 43.0 | 15.0 |
| **10** | 48.1 | 18.5 | 12.1 | 50.0 | 20.0 | 30.0 |

[a]% total weight does not sum to 100 due to fibre content.

### Experimental design
The study ran as two separate cohorts of 60 males each. Tissues from the first cohort were used to assess testis histology, testicular gene expression and testicular 8-OHdG DNA modification and males were culled after 15–19 weeks on diets. Males from each diet were systematically spread across this collection window to limit bias. Tissues from the second cohort were used to assess sperm motility, concentration, morphology, viability and ROS production, endocrinology and antioxidant activity of sperm and testes and males were culled after 16 weeks on diets. Body weight, body composition and reproductive organ weights were assessed in both cohorts.

### Animal husbandry
All procedures were reviewed and approved by the University of Sydney animal ethics committee (project number 2019/1610). Four-week-old male C57BL/6J (JAX strain code 000664) mice ($n = 120$) were acquired from the Animal Resources Centre (Murdoch, Australia). Upon arrival, males were housed in group cages of 3 animals, with random assignment by animal facility staff. Mice were housed with a 12 h light/dark cycle at 20–24 °C and 40–70% humidity. Males were allowed to acclimate for 3–4 days prior to being randomly allocated to an experimental diet ($n = 12$ males per diet across two cohorts with 6 males each). Diets were allocated in blocks (cages). No animals were excluded throughout the length of the study. Males received experimental diets for a minimum of 15 weeks and up to 19 weeks, and food and water were supplied *ad libitum* throughout the experimental period. Males were maintained in social housing groups of 3 animals per cage for the first 12 weeks (as required under animal ethics approval), and then moved to individual housing prior to metabolic assays being performed. Food intake was measured in the first cohort by weighing food in and out of individual cages over two 24 h periods at 16 and 20 weeks of age (i.e. after 11 and 15 weeks on diet). Bedding was changed at the start of intake measures and sifted for food crumbs to obtain as accurate measures as possible. The average of both measures was used in analyses. At sacrifice, animals were anaesthetised with sodium pentobarbital (100 mg/kg) and exsanguination by cardiac puncture was performed. Testes and seminal vesicles were removed and weighed immediately following sacrifice, fixed as detailed under histological analysis or snap frozen in liquid nitrogen and stored at −80 °C until further use.

### Body composition
Animals were weighed weekly and detailed body composition was measured after 10-13 weeks on diet by quantitative magnetic resonance using an EchoMRI-900-A130 (EchoMRI, Houston, USA).

### Sperm isolation and basic semen parameters
Immediately following sacrifice, sperm were isolated from $n = 6$ males per diet and used for all subsequent assessments. Both cauda epididymides were isolated and transferred to a Petri dish containing warmed 600 μL Ham's F10 medium (with 25 mM HEPES and 1 mM L-glutamine, catalogue number 12390-035, Thermo Fisher Scientific, Riverstone, Australia) supplemented with 1 mg/mL polyvinyl alcohol (87-90% hydrolyzed, average molecular weight 30,000–70,000), referred to subsequently as Ham's F10 + PVA. Epididymides were cut 5 times with a sterile scalpel blade and the tissue incubated for 15 min at 37 °C. Tissues were discarded and concentration of the isolated spermatozoa was determined using a Neubauer haemocytometer. Samples were subsequently diluted to $20 \times 10^6$ spermatozoa/mL with Ham's F10 + PVA and maintained at 37 °C.

Subjective motility was assessed in diluted samples within 90 mins of collection. A 5 μL sample was covered with a coverslip and immediately assessed under a phase contrast microscope using a ×10 objective. Motility was scored in increments of 5% by a single observer (TP). Morphology was assessed in eosin-nigrosin stained smears. A 5 μL

aliquot of the diluted sample was mixed with 5 μL of warmed eosin-nigrosin (final concentrations 0.25% w/v eosin, 2.5% w/v nigrosin) on a microscope slide and incubated for 30 s. The stained sample was gently smeared and allowed to air dry. Sperm morphology was assessed under a phase contrast microscope using a ×40 objective. A total of 200 spermatozoa were counted per slide and scored as normal or abnormal. Abnormal morphology included misshapen head, bent head, bent midpiece, bent tail and coiled tail.

## Flow cytometric assessment of sperm viability and ROS production

Viability was assessed using the LIVE/DEAD fixable red dead cell stain (Thermo Fisher Scientific) as per the manufacturer's directions. Briefly, a 50 μL aliquot of samples diluted to $20 \times 10^6$ spermatozoa/mL as described above was further diluted to $2 \times 10^6$ spermatozoa/mL with phosphate buffered saline (PBS) supplemented with 1 mg/mL polyvinyl alcohol, referred to subsequently as PBS + PVA. One microlitre of stain was added and samples were incubated at 37 °C for 30 min, then washed with 500 μL PBS + PVA (600x$g$, 2 min). The resultant pellet was fixed with 10% neutral buffered formalin for 15 min, washed with 1 mL PBS + PVA (600x$g$, 2 min) and stored at 4 °C in the dark until assessment (up to 4 days later).

Reactive oxygen species production was assessed using 2′,7′-dichlorofluorescein diacetate ($H_2DCFDA$). A 100 μL aliquot of samples diluted to $20 \times 10^6$ spermatozoa/mL as described above was stained (final concentration 5 μM) at 37 °C for 30 min. Excess stain was removed by centrifuging samples (600x$g$, 2 min) and resuspending in fresh Ham's F10 + PVA. Samples were assessed for baseline ROS production immediately following stain loading and after a further 1 h incubation period. At each time point, an aliquot was counterstained with propidium iodide (PI; final concentration 6 μM) for 5 min prior to assessment to discriminate the viable population.

Both the fixable viability and $H_2DCFDA$ stains were assessed on an LSRFortessa X-20 (Becton Dickinson, North Ryde, Australia), using FACSDiva software (v 9.0). Stains were excited by a 60 mW 488 nm laser and captured by a standard PMT trigon detector employing 610/10 BP (fixable viability stain, PI) and 525/50 BP ($H_2DCFDA$) filters. Voltage of the 525/50BP detector was standardised each experimental day using fluorescein isothiocyanate conjugated microparticles. Samples were acquired at a low flow rate, with a minimum of 10,000 events in the target population. Data analysis was performed with FlowJo software (v 10.7.2). As samples were dual stained with $H_2DCFDA$ and PI, single stained controls were used to create a compensation matrix for the experiment. All samples were gated firstly on the basis of forward and side scatter to isolate spermatozoa from debris, and subsequently on the basis of forward scatter area and height to isolate single cells. For the fixable viability stain, a 610/10 BP histogram was used to determine the proportion of viable (unstained) spermatozoa. For $H_2DCFDA$, the population of single cells was further gated based on a 610/10 BP histogram to discriminate the viable (PI negative) population and subsequently measure median 525/50 BP detector fluorescence. Examples of the gating strategy for each stain are shown in Supplementary figure 2.

## Histological analysis of testis

One testis per male ($n = 6$ per diet) was fixed for 24 h in 10% neutral buffered formalin, cut in half, transferred to 70% (v/v) ethanol and stored at 4 °C until processing. Utilising a tissue processer, the samples were dried by an ascending series of ethanol and paraffin embedded. Paraffin-embedded testis tissue was sectioned (5 μm) in the transverse plate with a rotary microtome. The tissue sections were lifted onto glass (Thermo Fisher Scientific). The slides were initially air-dried on a slide rack and were subsequently put into a 37 °C oven to dry overnight.

Slides were immersed in Harris' Haematoxylin for 2 min, washed in running distilled water 1 min, dipped 10x in 0.25% (v/v) acid alcohol, washed 1 min, immersed in Scott's blueing solution 30 sec and washed 2 min. Slides were then dehydrated in 70% (v/v) ethanol and counterstained in Eosin by 2 × 40 s immersions. Slides were further dehydrated by immersions in 95% (v/v) ethanol 2 × 1 min and then 100% (v/v) ethanol 2 × 1 min. Slides were then air dried and dipped in xylene. Slides were cover-slipped using dibutylphthalate polystyrene xylene mounting medium.

Slide imaging was performed using the Zeiss Zen (blue edition, v 3.4) software on an AxioScope-A1 microscope connected to an AxioCam-105 colour camera (Zeiss, North Ryde, Australia), using ×20 and ×40objectives. The analysis of testicular architecture between groups was performed on ImageJ freeware (v 2.1.0) by one operator (GA) who was blinded to treatment. Seminiferous tubule diameter and area, epithelium height and lumen diameter at the ×20 objective were measured using the measure tool in ImageJ. Johnsen scoring was carried out as per Johnsen[69], with a score of 10 indicating normal spermatogenesis and a score <3 indicating complete absence of any germ cells. Vacuolation within the epithelium was scored on a semi-quantitative scale of 1 to 5, with <1 (1), >5 (2), >10 (3), >15 (4), or >20 (5) vacuoles per tubule present.

## Endocrine analysis by LC-MS/MS

Whole blood was collected ($n = 6$ males per diet) by cardiac puncture into serum separator tubes and allowed to stand for up to 2 hours at room temperature. Samples were centrifuged (2000x$g$, 15 min) and the serum isolated and stored at −80 °C. Testosterone, dihydrotestosterone (DHT), esterone (E1), estradiol (E2), dehydroepiandrosterone (DHEA), 3α-androstanediol (3α-diol) and 3β-androstanediol (3β-diol) were assessed in thawed serum by LC-MS/MS as in ref. [70].

## Isolation of testicular RNA and DNA

One snap frozen testis per male ($n = 6$ males per diet, epididymis removed) was used to isolate RNA and DNA using a GenElute RNA/DNA/Protein Purification Plus Kit according to manufacturer's directions. Briefly, tissues were homogenised in SK buffer with β-mercaptoethanol using a TissueLyser (50 Hz, 1 min, repeated once) and a sonicating water bath (1 min). A standardised volume of lysate was loaded onto a gDNA purification column, washed, eluted and stored at −80 °C. RNA underwent on-column DNA digestion using 25 Kunitz RNA-free DNase (Qiagen, Doncaster, Australia). Digested DNA was removed, and the remaining RNA washed, eluted and stored at −80 °C. Concentrations of isolated DNA and RNA were determined spectrophotometrically using a NanoDrop 2000.

## Testicular gene expression by RT-qPCR

Isolated testicular RNA concentration was standardised using PCR grade water. Samples ($n = 6$ males per diet) were reverse transcribed using the high-capacity cDNA RT Kit (Thermo Fisher Scientific), with a total of 2 μg RNA, on an Eppendorf Nexus thermal cycler with manufacturer recommended cycle settings (25 °C, 10 min/37 °C, 120 min/85 °C, 5 min). A no-RT control was run, omitting the reverse transcription master mix. Generated cDNA was stored at −20 °C until use.

Quantitative PCR was performed using pre-designed KiCqStart primers or custom primers designed using NCBI Primer Blast (ncbi.nlm.nih.gov/tools/primer-blast) against the genes *Gpx1, Gpx4, Gss, Gsr, Nfe2l2, Adh5, Sod1, Sod2, Sod3, Cat, Hsd17b3, Star, Actb, RplpO* and *Tfrc*. Sequences of all primers are supplied in Supplementary table 1. Reaction parameters including annealing temperature and cDNA dilution were optimised for each primer set according to ref. [71].

qPCR reactions included 2 μL cDNA at the optimised dilution (1:10-1:60), 450 nM each of the forward and reverse primer, 5 μL PowerUp SYBR green 2x master mix (Thermo Fisher Scientific) and PCR grade

water to a total of 10 μL. All samples were assayed in duplicate. Samples were run on a LightCycler 480 II (Roche, Millers Point, Australia), with optimised cycle settings (UDG activation 50 °C, 2 min, polymerase activation 95 °C, 2 min, 40 cycles of denature 95 °C, 15 s, anneal at optimised temperature, 15 s, extension 72 °C, 1 min). Data were analysed using LightCycler 480 software (v 1.5.1.62, Roche). All expression data was standardised to reference genes (*Actb, RplpO, Tfrc*) and compared based on ΔΔCT.

### Testicular 8-OHdG DNA modification
Isolated testicular DNA (*n* = 5 males per diet) was assayed for oxidative modification using a colorimetric 8-hydroxy 2 deoxyguanosine (8-OHdG) ELISA kit (Abcam, Melbourne, Australia, cat # ab201734) according to the manufacturer's directions. Briefly, 4 μg of DNA was heat denatured (95 °C, 5 min) and treated with 20 units nuclease P1 (New England BioLabs, Notting Hill, Australia) at 37 °C for 5 min. Digested DNA was treated with alkaline phosphatase (0.01 units/μg of DNA, 37 °C, 30 min). 8-OHdG was detected using an anti-8-hydroxy 2 deoxyguanosine HRP-conjugated antibody (1:100 dilution) and TMB substrate, with absorbance measured at 450 nm using an Infinite 200 Pro plate reader (Tecan, Port Melbourne, Australia). 8-OHdG concentration was interpolated from a standard curve using GraphPad Prism software (v 9.0.1).

### Antioxidant activity of sperm and testes
Total antioxidant capacity (TAC) in both testes and sperm (*n* = 6 males per diet) were measured using a colorimetric assay kit based on reaction with a $Cu^{2+}$ reagent (Millipore Sigma) according to manufacturer's instruction. For the testis, isolated testicular tissue was homogenised in PBS, sonicated for 30 s and centrifuged (10,000x*g*, 2 min) to isolate supernatant. Protein concentration was measured by BCA and standardised to 0.1 mg/mL. For sperm, a total of 3 million spermatozoa were washed with PBS (600x*g*, 2 min) and the resulting pellet snap frozen and stored at −80 °C. The pellet was subsequently thawed on ice, probe sonicated 3 times (10 s on, 10 s off) and centrifuged (10,000x*g*, 2 min), and the supernatant retained. Samples were incubated for 90 min in the dark at room temperature with $Cu^{2+}$ reagent. Absorbance was measured at 570 nm using an Infinite M1000 Pro plate reader (Tecan).

Total glutathione concentration in testes (*n* = 6 males per diet) was measured using a fluorometric assay kit based on reaction with o-phthalaldehyde (BioVision, Milpitas, USA), according to manufacturer's directions. Briefly, 40 mg of frozen testis tissue was homogenised in assay buffer, sonicated for 10 s, and 60 μL of lysate was diluted with 20 μL chilled 6 N perchloric acid and centrifuged (13,000x*g*, 2 min) to isolate supernatant. Perchloric acid was precipitated with 20 μL chilled 6 N potassium hydroxide and removed by centrifugation (13,000x*g*, 2 min). Sample was diluted with assay buffer, treated with a reducing agent to convert GSSG to GSH, incubated 40 min at room temperature and fluorescence was measured at ex/em 340/420 nm using an Infinite M1000 Pro plate reader (Tecan).

### Statistical analysis
Data were analysed in R (v 4.1.0) using mixture models[72] (*mixexp* package) to account for interactive effects of dietary macronutrients. A total of 5 models, including a null model, were fitted for each experimental variable and tested for linear, quadratic and cubic effects of protein, fat and carbohydrate. Model selection was based on the lowest relative Akaike information criterion (AIC) as a measure of goodness of fit. For data visualisation, the selected predicted model was plotted as a right-angled mixture triangle response surface[73]. The 10 experimental diets are plotted in Supplementary figure 1 to indicate their relative positions within the response surface. AIC values and a summary of the selected model is provided for all variables in Supplementary data 1 and raw data for all outcome variables is given in Supplementary data 2. Correlation between parameters was assessed by Spearman rank correlation.

### Reporting summary
Further information on research design is available in the Nature Portfolio Reporting Summary linked to this article.

## Data availability
All data generated in this study are provided in the Supplementary Information/Source Data files. Source data are provided with this paper.

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

## Acknowledgements
This research was funded by a Challenge Programme Grant from the Novo Nordisk Foundation under grant agreement NNF18OC0033754 (R.B., S.J.S. and M.A.N.). The Novo Nordisk Foundation Center for Basic Metabolic Research is an independent research center at the University of Copenhagen, partially funded by an unrestricted donation from the Novo Nordisk Foundation (NNF18CC0034900, R.B.). Victoria Pye is acknowledged for her contribution to animal husbandry and technical assistance. The authors gratefully acknowledge the services of the ANZAC Research Institute andrology laboratory, the University of Sydney Charles Perkins Centre flow cytometry, histology and clinical imaging core facilities.

## Author contributions
A.J.C. and T. Pini contributed to experimental design, data collection and analysis, manuscript drafting, and critical revision. S.A. contributed to data collection and manuscript drafting. T. J. Pulpitel contributed to experimental design and data collection. G.A. contributed to data collection and manuscript drafting. A.M.S. contributed to data analysis and critical revision. H.N., T.F. and F.M. contributed to data collection. R.B., M.A.N. and S.J.S. contributed to experimental design, interpretation of results, and provided resources. All authors approved the final version of the manuscript.

## Competing interests
The authors declare no competing interests.
