## [Peer Review File · Nature Communications]

Male reproductive traits are differentially affected by dietary macronutrient balance but unrelated to adiposityREVIEWER COMMENTS

Reviewer #1 (Remarks to the Author):

The paper by Afrin et al. deals with reproductive parameters measured under 10 different macronutrient feeding conditions. Dietary fat had a positive influence on sperm motility and antioxidant capacity.

The reviewer is not an expert in the field of nutrient research. The way of displaying results mostly as a triangle-based graph was not very telling for someone not used to it. The manuscript lacked an explanation for a broader audience to transport the main message. As a consequence, the results should be presented in a better way.

As a detail, the way of assessing sperm motility by a single researcher by eye needs to be improved, as there are better methods described (line 115).

Reviewer #2 (Remarks to the Author):

The study by Crean and colleagues addressed the relation between macronutrient composition of diet and a series of reproductive traits in male mice. After testing ten isocaloric diets differing in their relative composition of protein, fat, and carbohydrates, the authors find that the macronutrient composition of diet has a variety of impacts on reproductive traits, including reproductive organ weights, semen parameters, testicular architecture, and testicular gene expression.

I think this paper is a very important contribution in at least two regards. First, the experimental design allows the identification of energy-independent effects of the macronutrient composition of diet. This is enormously important in animal nutrition experimentation since default dietary advice for humans is based on making isocaloric dietary substitutions while the overwhelming majority of animal experiments testing health effects of different diets does not make isocaloric comparisons, thus making it impossible to distinguish diet effects that are attributable to weight gain from dietary effects that are attributable to diet composition. This shift in experimental design is enormously important as it makes nutritional experimentation in animals immensely more relevant to humans, which is most certainly not the case to date. This alone, in my opinion, merits the publication of this paper in the most prominent venue possible. I very much look forward to seeing more work from this group using isocaloric diets to evaluate health effects of diet, as this is a major gap in the animal model to human translational research in nutrition.

A second important contribution of the paper are the findings themselves. Although there is a solid literature showing that obesity has a detrimental effect on male reproductive function both in animal models and in humans, whether diet can have energy-independent effects on male reproductive function is not as clear. Although there are observational studies in humans suggesting this may be the case, some cited by the authors, this question has not been addressed to date in animal experiments. Regardless of what the experiments would have shown, evaluating male reproductive outcomes in isocaloric comparison of diets differing in macronutrient composition is an important contribution to the literature.

These strengths notwithstanding, the manuscript has some important limitations as well as some minor issues. My major concerns are the following:

1. Although, as mentioned above, making isocaloric comparisons of diets differing in macronutrient content is extremely useful, some of the chosen experimental diets may not be particularly informative for human translation. One key issue are the ranges of macronutrient intake chosen for the experimental diets. Ranges of intake (as percent of energy) for total protein, total fat, and total carbohydrate were 7-50%, 15-60%, and 20-78%. While these intakes may provide interesting biological insights, they are completely irrelevant to humans. For example, although the observed range of intake for carbohydrates in the US general population resembles the experimental range

(~20-75% of energy as carbohydrates), this range is much wider than the Acceptable Macronutrient Distribution Range (AMDR) proposed by the National Academy of Medicine, Food and Nutrition Board (45-65%; NAS 2005). More concerning are the intake ranges for protein and fat which not only are much wider than the AMDR ranges but also much wider than intake ranges observed in the general population in the US and internationally (see, for example, PMID: PMC7326590). In hindsight, it may have been preferable to make fewer contrasts that in aggregate could allow the evaluation of high vs. low intake of specific macronutrients or dose-response relations (see, for example, the design of the experimental diets in PMID: PMC2763382). With this in mind, what is the point of testing diets with compositions that offer no translatable information to humans? Similarly, many of the effects identified, particularly those presented in Figures 4 and 5, appear to be limited to the edges of macronutrient distribution contrasts, and may therefore not have a direct mouse to human translation implications. I think it is important that the authors acknowledge these important limitations and discuss what to make of these findings and how could these inform the design of future studies in humans and animal models.

2. On the other hand, some of the effects identified appear to be primarily explained by variation in a single macronutrient regardless of variation in the other two (e.g. sperm motility appears to be mostly a function of protein intake [Fig 4a], lumen diameter appears to be mostly a function of carbohydrate intake [Fig 5b], testicular expression of Hsd17b3, Gpx1 and Gpx4 appear to be mostly a function of protein intake), suggesting that a simpler experimental design could have uncovered these effects. This raises the question of what is the utility of the complex design chosen by the authors, with ten experimental groups, and whether more efficient designs to address the same question may be preferable. Given that the design of the experimental diets may be one of the most important contributions of this paper to the field, I think devoting some space to questioning the rationale and efficiency of this method is more than warranted.

3. The description of the experimental diets is insufficient in my opinion. Describing the total macronutrient composition is not sufficient since this is the experimental group per se. Since it is well recognized that the relative intake of sub-classes of macronutrients is more important than that of total macronutrients across a broad range of health outcomes (particularly for cardio-metabolic health outcomes) in humans, it is essential to know not only the overall ranges of intake for total fat, protein and carbohydrates for each experimental group but also the composition of major subtypes of macronutrients. At minimum, I would like to know the distribution of intakes of major types of fats (saturated, cis-monounsaturated, n-3 cis-polyunsaturated, n-6 cis-polyunsaturated and trans-unsaturated), a better description of carbohydrate intake in terms of some measure of carbohydrate quality (e.g. total sugar, added sugars, total soluble and insoluble fiber), and the relative use of animal vs vegetable sources of protein and/or a description of amino acid composition of each of the experimental diets. It would also be important to describe intakes of micronutrients, particularly fat soluble vitamins, across the different experimental groups. These additional information is particularly relevant since the interpretation of the findings in the discussion delves into relative intakes of sub-types of dietary fat when there is no evidence that the experimental diets differed in sub-types of dietary fat. In fact, the very cursory description provided suggested that composition of sub-types of fat was kept constant. Please back up your claims.

4. The authors are to be commended for the extensive reproductive phenotype characterization of the experimental animals. Nevertheless, it is increasingly recognized that standard semen parameters and even functional measures of semen quality are only very weakly related to fertility and only as it relates to variation in the low end of the distribution of these markers. Was there any attempt to mate these animals and test their ability to sire offspring? At the same time, there are well described differences in semen parameters and reproductive efficiency between rodents and humans (if not immediately obvious, humans are very bad at making sperm and reproducing and even marginal rodents would make good humans, in terms of reproductive capacity). Knowing that, does the lack of association with some semen parameters have any implications for humans? Also, what are the implications of the individual findings, as a whole on reproductive performance? Ultimately, the purpose of studying these proxy phenotypes is not because they are important by their own merits but because they shed light on fertility. Those implications are missing from the discussion.

I have also two minor concerns regarding the methods:

5. Was the assessment of abnormal morphology hierarchical (i.e. head defects are scored first and if defects are found no other defects are documented; mid-piece defects are scored next and if defects are found no other defects are documented; tail defects are scored last and only documented in sperm without head or mid-piece defects) or were all morphological defects scored? If all defects were scored, were there any differential effects for head, mid-piece and tail defects?

6. Please confirm whether testicular gene expression studies were conducted in the testis only or whether the epididymis was included in the testis gene expression studies.

Reviewer #3 (Remarks to the Author):

The presented research expands the complement of studies related to nutritional geometry to include male mouse reproductive/fertility related outcomes. There are multiple positives regarding the study in its design and descriptions, which addresses research questions with a unique approach. At present, the strengths are counterbalanced with several concerns and limitations that are not fully addressed in the design or text in its current form. These are noted below for reference.

Overview/Abstract/Introduction

The pretext of the study, as presented by the authors, with obesity-related male infertility and clinical recommendations to lose weight prior to attempting to conceive does not align with the study design as executed which started with very young, healthy animals and did not begin with an obese or overweight state. Thus, any infertility related to obesity is not possible to study with the presented design, in that direct comparisons made with male fertility impairment followed by dietary intervention for weight loss were not included. The authors' reframed the question at the end of the introduction to focus more specifically on the effects of macronutrient balance/proportion on male reproductive aspects, but whether similar observations to those presented with the various dietary compositions would also occur from dietary interventions after the prior establishment of obesity is not tested.

Methods

Overall, the methods are detailed with relevant information to promote RRT in the study. There are a few cases where more specific information would be helpful.

For instance, details of the design and animal randomization/allocations could benefit from further description. One reading of the methods suggests the group housing for the majority of the study duration would support the need for a statistical model that incorporates co-housing, which was not clear from the statistical analysis descriptions. Similarly, many outcomes were for n=6/diet treatment, but unclear as to whether these represented n=2 cages of animals from the same cohort run, animals from different cohorts (n=2 total cohorts are indicated), or some combination thereof?

A minimum study duration of '15 weeks and up to 19 weeks' is noted, but it is unclear from the manuscript whether any of the reported outcomes would be expected to change significantly during this period of time where variability in the collection of measures could influence analyses respectively? How were animals from different diet treatments selected for collections and completed within this 4-week range?

QMR via an EchoMRI-900 is noted as utilized for body composition assessment. Details are needed regarding the equipment specifications as the small size of the animals (appear to be 20-30+ grams) in the present study raise questions about the validity of the noted QMR system for animals up to 900 grams if the system had no additional hardware of scanning sequence parameters.

Throughout the methods description of outcomes, it is not clear if specific animals were used for all outcomes or if different animals within a diet treatment contributed to different outcomes (e.g. how do the n=6 for sperm isolation related to the n=6 for histological analysis versus the n=6 for whole blood collection outcomes relate to the n=12 total)?

Were any animals missing measures for any outcomes and/or timepoints related to the study?

It is noted that cellulose was utilized to balance the dietary compositions for calorie density, but

does not appear to have been incorporated in the statistical analyses? Reporting the specific amounts of cellulose added to each diet in Table I would also be informative. Additionally, incorporation of cellulose dietary content could be added to the assessments unless there is a biological reason there would be no potential influence on any of the reported study outcomes. While there are many strengths of the nutritional geometry approach, there could also be some amount of compensation through intake differences among animals and diet treatment groups. Was intake measured and if so, could it be reported. If intake was not measured, an explanation for why not and/or implications for potential differences in intake versus dietary macronutrient composition and the outcomes reported should be considered and discussed.

Results

Based on the reported values for body weight and composition (with the extremely small amount of fat mass reported [~ 5 grams or less] raising concerns from the QMR precision and accuracy as noted in the Methods section), it appears likely none of the groups achieved an 'obese' state which relates to the pretext for the overall study.

The use of % body fat, rather than a model with fat and lean mass incorporated as co-variables, may raise some issue about the overall interpretation of results for specific outcomes as noted in various sections. The uncertainty regarding the body composition could have manifold impacts on the overall study interpretations.

Discussion

The majority of the primary outcomes appear to be by-in-large unaffected by the dietary treatments tested. While there are some sub-analyses that identify specific differences between or among diet groups, those do not stand out as explanatory for the primary hypothesis presented regarding obesity-related infertility and dietary interventions to improve reproductive outcomes. As noted by the authors, there are multiple published studies which relate different dietary ingredients/amounts to reproductive outcomes, impacting the overall novelty of findings here despite the systematic investigation performed.

Some of the discussion about dietary energy may be relevant, but without additional information about dietary feeding/intake amounts, the dietary 'energy density' per se may be only part of the story presented.

Although there was a reported range of fat masses observed, do any of the groups achieve a status of 'obesity' as defined by animal model research?

The discussion of published studies and tested dietary ingredients versus the macronutrient manipulation while maintaining the same ingredient list is a strength of the present design, but leave primarily speculation as potential contributors of observed differences in outcomes given the lack of direct tests of standard high-fat diet comparators with the reported strain for the study ages/conditions (would have been a helpful control comparison).

Quite interestingly, multiple of the primary outcomes reported could be measured in clinical studies specific to the model of dietary interventions post-obesity establishment.

The final paragraph is a bit confusing in that the study seems to not recommend a single dietary composition, but the authors focus on the need for research into diet to understand specific contexts, etc.

Figures

N's for each figure and subpanel would be helpful for readers.

The significance of the correlations among multiple outcomes which are presented despite no significant differences in primary outcome by diet treatment assessment is not clear as to the relevance and explanatory contribution (this relates to both the text and figures).

Reviewer #1 (Remarks to the Author):

The paper by Afrin et al. deals with reproductive parameters measured under 10 different macronutrient feeding conditions. Dietary fat has a positive influence on sperm motility and antioxidant capacity.

1. The reviewer is not an expert in the field of nutrient research. The way of displaying results mostly as a triangle-based graph was not very telling for someone not used to it. The manuscript lacked an explanation for a broader audience to transport the main message. As a consequence, the results should be presented in a better way.

We appreciate that the right-angle mixture triangle (RMT) format for visualisation of results is novel for many readers, but it is not the appropriate way to present data from a proportions-based mixture design (see Raubenheimer, 2011). The other valid alternative is to use a simplex triangle rather than an RMT (Ruohonen & Kettunen, 2004), but these are far less intuitive for the reader than RMTs. Presenting 10 treatments in a traditional graph (e.g. bar graph) cannot allow the reader to interpret the effects of increasing a particular macronutrient in relation to others within the mixture and does not accurately represent the structure and design of the experiment and analysis. To improve accessibility of our dataset, we have now included all raw data for every outcome variable as a supplementary file (Supplementary table 3). We have also added an explanation of how to interpret the graphs in the caption of Figure 1.

2. As a detail, the way of assessing sperm motility by a single researcher by eye needs to be improved, as there are better methods described (line 115).

While there are alternative means of assessing sperm motility, particularly computer-assisted sperm assessment, subjective motility assessment remains a valid means of analysis, as described in the WHO laboratory manual for the examination and processing of human semen, 6th edition (2021). To limit any potential bias, all samples were assessed by one observer.

Reviewer #2 (Remarks to the Author):

The study by Crean and colleagues addressed the relation between macronutrient composition of diet and a series of reproductive traits in male mice. After testing ten isocaloric diets differing in their relative composition of protein, fat, and carbohydrates, the authors find that the macronutrient composition of diet has a variety of impacts on reproductive traits, including reproductive organ weights, semen parameters, testicular architecture, and testicular gene expression.

I think this paper is a very important contribution in at least two regards. First, the experimental design allows the identification of energy-independent effects of the macronutrient composition of diet. This is enormously important in animal nutrition experimentation since default dietary advice for humans is based on making isocaloric dietary substitutions while the overwhelming majority of animal experiments testing health effects of different diets does not make isocaloric comparisons, thus making it impossible to distinguish diet effects that are attributable to weight gain from dietary effects that are attributable to diet composition. This shift in experimental design is enormously important as it makes nutritional experimentation in animals immensely more relevant to humans, which is most certainly not the case to date. This alone, in my opinion, merits the publication of this paper in the most prominent venue possible. I very much look forward to seeing more work from this group using isocaloric diets to evaluate health effects of diet, as this is a major gap in the animal

model to human translational research in nutrition.

A second important contribution of the paper are the findings themselves. Although there is a solid literature showing that obesity has a detrimental effect on male reproductive function both in animal models and in humans, whether diet can have energy-independent effects on male reproductive function is not as clear. Although there are observational studies in humans suggesting this may be the case, some cited by the authors, this question has not been addressed to date in animal experiments. Regardless of what the experiments would have shown, evaluating male reproductive outcomes in isocaloric comparison of diets differing in macronutrient composition is an important contribution to the literature.

These strengths notwithstanding, the manuscript has some important limitations as well as some minor issues. My major concerns are the following:

1. Although, as mentioned above, making isocaloric comparisons of diets differing in macronutrient content is extremely useful, some of chosen experimental diets may not be particularly informative for human translation. One key issue are the ranges of macronutrient intake chosen for the experimental diets. Ranges of intake (as percent of energy) for total protein, total fat and total carbohydrate were 7-50%, 15-60% and 20-78%. While these intakes may provide interesting biological insights, they are completely irrelevant to humans. For example, although the observed range of intake for carbohydrates in the US general population resembles the experimental range (~20-75% of energy as carbohydrates), this range is much wider than the Acceptable Macronutrient Distribution Range (AMDR) proposed by the National Academy of Medicine, Food and Nutrition Board (45-65%; NAS 2005). More concerning are the intake ranges for protein and fat which not only are much wider than the AMDR ranges but also much wider than intake ranges observed in the general population in the US and internationally (see, for example, PMID: PMC7326590). In hindsight, it may have been preferable to make fewer contrasts that in aggregate could allow the evaluation of high vs. low intake of specific macronutrients or dose-response relations (see, for example, the design of the experimental diets in PMID: PMC2763382). With this in mind, what is the point of testing diets with compositions that offer no translatable information to humans? Similarly, many of the effects identified, particularly those presented in Figures 4 and 5, appear to be limited to the edges of macronutrient distribution contrasts, and may therefore not have a direct mouse to human translation implications. I think it is important that the authors acknowledge these important limitations and discuss what to make of these findings and how could these inform the design of future studies in humans and animal models.

Thank you for your enthusiasm and support for our study and for careful consideration and thoughtful review of our experimental design. We have greatly expanded our discussion of the strengths and limitations of our study in response to your suggestions (L485-500). The dietary ranges selected for this study were based on results of previous studies examining health effects of macronutrient balance in C57BL6/6J mice (e.g., Solon-Biet et al., 2014; Le Couteur et al., 2021). As acknowledged by the reviewer, this is the first time such a design has been used to examine male reproductive traits. Hence, we designed the study to cover the full range of physiologically viable nutrient intakes to be as comprehensive as possible and to allow interpolation of responses in realistic regions within the full range tested. We have taken exactly this approach previously for

human data in relation to energy intake, obesity and longevity (Raubenheimer et al., 2015; Raubenheimer & Simpson, 2016), imposing the AMDR upon response surfaces derived from a broader range of macronutrient mixtures. This design allowed us to clearly demonstrate both linear and more complex main and interactive relationships that would not otherwise be apparent in a pairwise comparison or a design that was more limited in its coverage of nutrient space (e.g. low fat vs high fat).

2. On the other hand, some of the effects identified appear to be primarily explained by variation in a single macronutrient regardless of variation in the other two (e.g. sperm motility appears to be mostly a function of protein intake [Fig 4a], lumen diameter appears to be mostly a function of carbohydrate intake [Fig 5b], testicular expression of Hsd17b3, Gpx1 and Gpx4 appear to be mostly a function of protein intake), suggesting that a simpler experimental design could have uncovered these effects. This raises the question of what is the utility of the complex design chosen by the authors, with ten experimental groups, and whether more efficient designs to address the same question may be preferable. Given that the design of the experimental diets may be one of the most important contributions of this paper to the field, I think devoting some space to questioning the rationale and efficiency of this method is more than warranted.

Yes, we were surprised (and pleased) that some of the response traits showed clear linear effects of individual macronutrients, simplifying follow-up experiments. This could not have been anticipated a priori, however. It is much more compelling when simple relationships emerge from a design from which far more complex interactions could have appeared. Also, the design allowed us to identify traits that responded to different nutrients (fat and carbs in this instance), even when responses were linear and single nutrient. Post hoc, our findings certainly allow simpler experimental designs to be pursued at the next stage – but this could not have been appreciated in advance. Therein lies the power of nutritional geometry – it allows the complexity of nutrient interactions to be tamed sequentially. Nutritional geometry provides an essential first step, uncovering both individual and interactive macronutrient effects across the full nutritional landscape. These data can then be used to generate hypotheses for future studies employing simpler designs thanks to the comprehensive nature of this foundational study. We appreciate that this approach is complex and by its nature places limitations on the experimental design, which we have now pointed to in the discussion (L485-500). Many of the complexities we address, notably around interdependent effects of nutrients in diets, are obscured and overlooked in more conventional single nutrient diet designs – they are inescapable properties of mixtures.

3. The description of the experimental diets is insufficient in my opinion. Describing the total macronutrient composition is not sufficient since this is the experimental group per se. Since it is well recognized that the relative intake of sub-classes of macronutrients is more important than that of total macronutrients across a broad range of health outcomes (particularly for cardio-metabolic health outcomes) in humans, it is essential to know not only the overall ranges of intake for total fat, protein and carbohydrates for each experimental group but also the composition of major subtypes of macronutrients. At minimum, I would like to know the distribution of intakes of major types of fats (saturated, cis-monounsaturated, n-3 cis-polyunsaturated, n-6 cis-polyunsaturated and trans-unsaturated), a better description of carbohydrate intake in terms of some measure of carbohydrate quality (e.g. total sugar, added sugars, total soluble and insoluble fiber), and the relative use of animal vs vegetable sources of protein and/or a description of amino acid composition of each of the experimental diets. It would also be important to describe

intakes of micronutrients, particularly fat soluble vitamins, across the different experimental groups. This additional information is particularly relevant since the interpretation of the findings in the discussion delves into relative intakes of sub-types of dietary fat when there is no evidence that the experimental diets differed in sub-types of dietary fat. In fact, the very cursory description provided suggested that composition of sub-types of fat was kept constant. Please back up your claims.

Detailed descriptions of each diet, including the amount of each ingredient and amino acid, fatty acid and micronutrient compositions as fed are now included in Supplementary file 1.

We can confirm, while the overall levels of macronutrients changed between diets, the ratios of macronutrient sub-types were kept near constant– this has been detailed in the methods (L81-82) for carbohydrates, omega 3/6 fatty acids and saturated fats.

4. The authors are to be commended for the extensive reproductive phenotype characterization of the experimental animals. Nevertheless, it is increasingly recognized that standard semen parameters and even functional measures of semen quality are only very weakly related to fertility and only as it relates to variation in the low end of the distribution of these markers. Was there any attempt to mate these animals and test their ability to sire offspring? At the same time, there are well described differences in semen parameters and reproductive efficiency between rodents and humans (if not immediately obvious, humans are very bad at making sperm and reproducing and even marginal rodents would make good humans, in terms of reproductive capacity). Knowing that, does the lack of association with some semen parameters have any implications for humans? Also, what are the implications of the individual findings, as a whole on reproductive performance? Ultimately, the purpose of studying these proxy phenotypes is not because they are important by their own merits but because they shed light on fertility. Those implications are missing from the discussion.

Males in this study were mated to females to generate offspring for a follow up study, however the method of mating was not an accurate means of assessing fertility (i.e., we employed multiple matings per male and uncontrolled exposure time to females). Overall, these matings did not show any males with complete infertility, therefore a more robust approach would be required to detect any subtle differences in fertility, if they exist.

The mouse v human reproductive efficiency is certainly a valid point. Given the overall poorer 'baseline' reproductive parameters observed in human males and the multifaceted environmental/lifestyle factors negatively impacting male fertility, we would suggest that the observed effects may be more pronounced, or at least more problematic in humans.

Individual reproductive outcomes (e.g. sperm motility, morphology etc) are linked to fertility, but are not necessarily accurately predictive of fertility outcomes (e.g. pregnancy rate, time to successful conception, litter size). For semen parameters, there are clearly established cut offs correlated to fertility outcomes used in human reproductive medicine to predict male fertility. No such cut offs exist for mice, however in using mice as a model, we sought to establish whether any of these parameters might be altered by diet, with the view that if such a change also occurs in humans, it could bring men below the clinically accepted 'normal' cut offs.

These points have now been addressed in the discussion (L469-484).

I have also two minor concerns regarding the methods:

5. Was the assessment of abnormal morphology hierarchical (i.e. head defects are scored first and if defects are found no other defects are documented; mid-piece defects are scored next and if defects are found no other defects are documented; tail defects are scored last and only documented in sperm without head or mid-piece defects) or were all morphological defects scored? If all defects were scored, were there any differential effects for head, mid-piece and tail defects?

Individual sperm were scored as abnormal based on any abnormality within the sperm head or tail. Although they were not individually recorded, there was no bias towards head or tail abnormalities observed during analysis.

6. Please confirm whether testicular gene expression studies were conducted in the testis only or whether the epididymis was included in the testis gene expression studies.

qPCR was performed on testicular tissue only. This has been clarified in the methods (L207).

Reviewer #3 (Remarks to the Author):

The presented research expands the complement of studies related to nutritional geometry to include male mouse reproductive/fertility related outcomes. There are multiple positives regarding the study in its design and descriptions, which addresses research questions with a unique approach. At present, the strengths are counterbalanced with several concerns and limitations that are not fully addressed in the design or text in its current form. These are noted below for reference.

Overview/Abstract/Introduction

The pretext of the study, as presented by the authors, with obesity-related male infertility and clinical recommendations to lose weight prior to attempting to conceive does not align with the study design as executed which started with very young, healthy animals and did not begin with an obese or overweight state. Thus, any infertility related to obesity is not possible to study with the presented design, in that direct comparisons made with male fertility impairment followed by dietary intervention for weight loss were not included. The authors' reframed the question at the end of the introduction to focus more specifically on the effects of macronutrient balance/proportion on male reproductive aspects, but whether similar observations to those presented with the various dietary compositions would also occur from dietary interventions after the prior establishment of obesity is not tested.

We have revised the Abstract and Introduction to de-emphasise obesity as suggested. However, although our study does indeed examine nutrient effects as opposed to obesity per se, we believe the focus on obesity in the Introduction is appropriate given that it is the focus of the vast majority of previous publications in this field of study.

Methods

Overall, the methods are detailed with relevant information to promote RRT in the study. There are a few cases where more specific information would be helpful. For instance, details of the design and animal randomization/allocations could benefit from further description.

1. One reading of the methods suggests the group housing for the majority of the study duration would support the need for a statistical model that incorporates co-housing, which was not clear from the statistical analysis descriptions.

The mixture-format of the experiment necessitates analysis using mixture models (also known as Scheffe's polynomials). We analysed our data using the mixExp package implementation, which includes 5 models that allow for a series of null, through linear through non-linear effects. Unfortunately, this package, does not allow for the inclusion of random effects, as would be required to correct for cage effects. We can however fit the simplest linear mixture model outside of the package to include a random effect. We have now implemented this analysis for all outcomes in the study, and found that the regression coefficients from the mixture model with and without the cage effect are all within ~2% of one another. Hence, including any cage effect does not alter the response to the treatments (see below).

With regards to the analysis in the main-text, we are forced to select between (1) a model that accounts for the cage effect, or (2) models that allow for non-linear responses to the experimental design. In our experience, accounting for non-linear responses in nutritional studies is important. Coupled with the below analysis indicating cage effects are unlikely to alter the estimated response to the treatments, we prefer the analysis accounting for non-linearity.

2. Similarly, many outcomes were for n=6/diet treatment, but unclear as to whether these represented n=2 cages of animals from the same cohort run, animals from different cohorts (n=2 total cohorts are indicated), or some combination thereof?

This has now been clarified under the 'experimental design' method subsection (L88-95), indicating which animals were used for each outcome measure.

3. A minimum study duration of '15 weeks and up to 19 weeks' is noted, but it is unclear from the manuscript whether any of the reported outcomes would be expected to change significantly during this period of time where variability in the collection of measures could influence analyses respectively? How were animals from different diet treatments selected for collections and completed within this 4-week range?

The two cohorts ran for different lengths of time – the first cohort for 15-19 weeks and the second for 16 weeks. For the first cohort, diets were systematically spread across collection window of 15-19 weeks to ensure no bias in collections. This has now been clarified in the methods (L 88-95).

4. QMR via an EchoMRI-900 is noted as utilized for body composition assessment. Details are needed regarding the equipment specifications as the small size of the animals (appear to be 20-30+ grams) in the present study raise questions about the validity of the noted QMR

system for animals up to 900 grams if the system had no additional hardware of scanning sequence parameters.

QMR was performed with an EchoMRI-900-A130 (i.e. with an A130 adapter, suitable for animals up to 130g). This has been clarified in the methods (L119).

5. Throughout the methods description of outcomes, it is not clear if specific animals were used for all outcomes or if different animals within a diet treatment contributed to different outcomes (e.g. how do the n=6 for sperm isolation related to the n=6 for histological analysis versus the n=6 for whole blood collection outcomes relate to the n=12 total)? Were any animals missing measures for any outcomes and/or timepoints related to the study?

This has now been clarified under the 'experimental design' method subsection (L88-95), indicating which animals were used for each outcome measure. In addition, the number of animals per outcome measure has been indicated throughout the methods and in figure legends.

6. It is noted that cellulose was utilized to balance the dietary compositions for calorie density, but does not appear to have been incorporated in the statistical analyses? Reporting the specific amounts of cellulose added to each diet in Table I would also be informative. Additionally, incorporation of cellulose dietary content could be added to the assessments unless there is a biological reason there would be no potential influence on any of the reported study outcomes.

Detailed descriptions of each diet, with the relative weight of each ingredient (including cellulose), are now included in Supplementary file 1. In addition, cellulose content of diets has been added to Table 1. While insoluble dietary fibre does not have any significant nutritional value, we appreciate that this component necessarily differed across diets to dilute calories. While this makes it difficult to separate the effects of dietary fat and insoluble dietary fibre (now included as a limitation in the discussion L485-495), we felt that it was most important to control the effects of calories rather than the level of insoluble dietary fibre. It is important to note that, because of the nature of mixtures, it is impossible to break the correlation between either fat and cellulose (in a design such as ours here, which keeps energy density constant), or between fat and energy (in a design that maintains dry mass relationships between macronutrients but therefore allows energy density to vary. We have employed designs which do both (e.g., Solon-Biet et al., 2014, 2015) but this requires multiplying the number of dietary treatments by each energy density tested. Here, our aim was to explore macronutrient ratios at constant energy density.

7. While there are many strengths of the nutritional geometry approach, there could also be some amount of compensation through intake differences among animals and diet treatment groups. Was intake measured and if so, could it be reported. If intake was not measured, an explanation for why not and/or implications for potential differences in intake versus dietary macronutrient composition and the outcomes reported should be considered and discussed.

Indeed and we have reported this previously (e.g., Solon-Biet et al., 2014, 2015). Intake was measured only in the one cohort of males – these data have now been added to the methods (L108-112), results (L301-302) and Figure 1(D). Consideration of intake versus diet composition have now been added to the discussion (L494-498).

Results

8. Based on the reported values for body weight and composition (with the extremely small amount of fat mass reported [~ 5 grams or less] raising concerns from the QMR precision and accuracy as noted in the Methods section), it appears likely none of the groups achieved an 'obese' state which relates to the pretext for the overall study.

Values plotted on surfaces are modelled values only and do not represent the actual minimum and maximum raw values. To address this, we have now included all raw data for every outcome variable as a supplementary file (Supplementary table 3). In the case of fat mass, the maximum observed fat mass/percentage was 12.85g/33%, which falls within the range of western diet induced obesity studies. Overall, the goal of the study was not to induce obesity, but to investigate the impacts of different dietary macronutrient ratios, some of which may lead to obesity. As we have shown previously (Solon-Biet et al., 2014, 2015), this allows obesity and other metabolic or reproductive outcomes to be dissociated in response surface plots.

9. The use of % body fat, rather than a model with fat and lean mass incorporated as co-variates, may raise some issue about the overall interpretation of results for specific outcomes as noted in various sections. The uncertainty regarding the body composition could have manifold impacts on the overall study interpretations.

Please see response above in regard to incorporation of co-variates in the statistical approach used. However, % body fat was not significantly correlated to other measures as demonstrated in Figure 2.

Discussion

The majority of the primary outcomes appear to be by-in-large unaffected by the dietary treatments tested. While there are some sub-analyses that identify specific differences between or among diet groups, those do not stand out as explanatory for the primary hypothesis presented regarding obesity-related infertility and dietary interventions to improve reproductive outcomes. As noted by the authors, there are multiple published studies which relate different dietary ingredients/amounts to reproductive outcomes, impacting the overall novelty of findings here despite the systematic investigation performed. Some of the discussion about dietary energy may be relevant, but without additional information about dietary feeding/intake amounts, the dietary 'energy density' per se may be only part of the story presented.

10. Although there was a reported range of fat masses observed, do any of the groups achieve a status of 'obesity' as defined by animal model research?

There is no widely accepted definition of obesity in research using mouse models – most publications simply refer to changes in body weight or fat mass compared to the control group, with most denoting a significant difference as evidence of obesity (as described in de Moura e Dias et al 2021). This has now been addressed in the discussion (L513-516).

11. The discussion of published studies and tested dietary ingredients versus the macronutrient manipulation while maintaining the same ingredient list is a strength of the present design, but leave primarily speculation as potential contributors of observed differences in outcomes given the lack of direct tests of standard high-fat diet comparators with the reported strain for the study ages/conditions (would have been a helpful control comparison).

While we appreciate that a direct comparison to a western diet is a more traditional approach, we feel that the comparison would not have provided much insight. As the study was designed to use

isocaloric diets, the inclusion of one non-isocaloric diet with a different macronutrient ratio presents the same concerns as the traditional control vs western diet approach of not comparing apples to apples. Including a non-isocaloric version of each diet would be another option, however this would have doubled the number of treatments and made many of the assessments unfeasible.

12. Quite interestingly, multiple of the primary outcomes reported could be measured in clinical studies specific to the model of dietary interventions post-obesity establishment.
The final paragraph is a bit confusing in that the study seems to not recommend a single dietary composition, but the authors focus on the need for research into diet to understand specific contexts, etc.

The final paragraph has now been updated (L523-534).

Figures

13. N's for each figure and subpanel would be helpful for readers.

n values have now been added in the figure legend for each figure.

14. The significance of the correlations among multiple outcomes which are presented despite no significant differences in primary outcome by diet treatment assessment is not clear as to the relevance and explanatory contribution (this relates to both the text and figures).

Correlations between body composition and reproductive parameters are shown in Figure 2, as a significant point within the discussion is the relationship between adiposity and reproductive outcomes. In addition, we feel it is of interest to demonstrate outcome measures which were correlated, regardless of whether they differed with diet, as this establishes whether the findings follow expected trends relative to other studies.

References

- de Moura e Dias, M. *et al.* Diet-induced obesity in animal models: Points to consider and influence on metabolic markers. *Diabetol. Metab. Syndr.* **13**, 32 (2021).
- Le Couteur DG, Solon-Biet SM, Parker BL, Pulpitel T, Brandon AE, Hunt NJ, Wali JA, Gokarn R, Senior AM, Cooney GJ, Raubenheimer D. Nutritional reprogramming of mouse liver proteome is dampened by metformin, resveratrol, and rapamycin. *Cell Metab.* **33**. 2367-79 (2021).
- Raubenheimer, D. Toward a quantitative nutritional ecology: The right-angled mixture triangle. *Ecol. Monogr.* **81**, 407-427 (2011).
- Raubenheimer, D., Machovsky-Capuska, G., Gosby, A., & Simpson, S. Nutritional ecology of obesity: From humans to companion animals. *Br. J. Nutr.* **113**(S1), S26-S39 (2015).
- Raubenheimer, D. & Simpson, S. J. Nutritional ecology and human health. *Annu. Rev. Nutr.* **36**, 603-626 (2016).
- Ruohonen K, Kettunen J. Effective experimental designs for optimizing fish feeds. *Aquac. Nutr.* **10**. 145-51 (2004).
- Solon-Biet, SM. *et al.* The ratio of macronutrients, not caloric intake, dictates cardiometabolic health, aging, and longevity in ad libitum-fed mice. *Cell Metab.* **19**, 418-430 (2014)

Solon-Biet, S. M. *et al.* Macronutrient balance, reproductive function, and lifespan in aging mice. *Proc. Natl. Acad. Sci. USA* **112**, 3481-3486 (2015)

WHO laboratory manual for the examination and processing of human semen, sixth edition. Geneva: World Health Organization; 2021.

REVIEWERS' COMMENTS

Reviewer #1 (Remarks to the Author):

My points have been addressed. I have no further comments

Reviewer #2 (Remarks to the Author):

The authors have satisfactorily addressed all my concerns. It was a pleasure to review this paper.

Reviewer #3 (Remarks to the Author):

The authors' responsiveness and thoroughness in addressing the points raised during the initial review are appreciated, and the updates to the manuscript are favorably received.

One minor comment arises in response to the updated Discussion, where the first sentence (line 373) states "that male reproductive physiology is principally under the influence of dietary macronutrient ratios...". The study design and results support that some of the measured aspects of male reproductive physiology are significantly influenced by dietary macronutrient ratios, but 'principally' may be overreaching as there are multiple reproductive outcome measures that were not significantly different among the groups and the rigor of the study conditions support a more narrowly defined conclusion within the strain, dietary ingredients, duration of feeding, ages of assessment, physical activity interactions, etc. as tested - while highlighting potential hypotheses about these broader concepts which could be more fully investigated in other studies.

Similarly, the 'running title' could be edited to reflect the measures performed - reproductive traits or some other phrase rather than 'reproduction' which appears to have not been directly measured.

The final comment from reviewer 3:

The authors' responsiveness and thoroughness in addressing the points raised during the initial review are appreciated, and the updates to the manuscript are favorably received.

One minor comment arises in response to the updated Discussion, where the first sentence (line 373) states "that male reproductive physiology is principally under the influence of dietary macronutrient ratios...". The study design and results support that some of the measured aspects of male reproductive physiology are significantly influenced by dietary macronutrient ratios, but 'principally' may overreaching as there are multiple reproductive outcome measures that were not significantly different among the groups and the rigor of the study conditions support a more narrowly defined conclusion within the strain, dietary ingredients, duration of feeding, ages of assessment, physical activity interactions, etc. as tested - while highlighting potential hypotheses about these broader concept which could be more fully investigated in other studies.

Similarly, the 'running title' could be edited to reflect the measures performed – reproductive traits or some other phrase rather than 'reproduction' which appears to have not been directly measured.

In response, we have edited the first sentence of the discussion (now line 157) and the running title to give a broader focus.